# Phenotype of White Sika Deer Due to SCF Gene Structural Variation

**DOI:** 10.3390/genes14051035

**Published:** 2023-05-02

**Authors:** Xu Chen, Shiwu Dong, Xin Liu, Ning Ding, Xiumei Xing

**Affiliations:** 1State Key Laboratory for Molecular Biology of Special Economic Animals, Institute of Special Animal and Plant Sciences of Chinese Academy of Agricultural Science, Changchun 130112, China; 2College of Wildlife and Protected Area, Northeast Forestry University, Harbin 150040, China

**Keywords:** SCF, sika deer, whole-genome sequencing, melanin

## Abstract

Breeding ornamental white sika deer is a new notion that can be used to broaden the sika deer industry However, it is very rare for other coat phenotypes to occur, especially white (apart from albinism), due to the genetic stability and homogeneity of its coat color phenotype, making it difficult to breed white sika deer between species. We found a white sika deer and sequenced its whole genome. Then, the clean data obtained were analyzed on the basis of gene frequency, and a cluster of coat color candidate genes containing 92 coat color genes, one SV (structure variation), and five nonsynonymous SNPs (single nucleotide polymorphisms) was located. We also discovered a lack of melanocytes in the skin tissue of the white sika deer through histological examination, initially proving that the white phenotype of sika deer is caused by a 10.099 kb fragment deletion of the *SCF* gene(stem cell factor). By designing SCF-specific primers to detect genotypes of family members of the white sika deer, and then combining them with their phenotypes, we found that the genotype of the white sika deer is SCF^789^/SCF^789^, whereas that of individuals with white patches on their faces is SCF^789^/SCF^1–9^. All these results showed that the *SCF* gene plays an important role in the development of melanocytes in sika deer and is responsible for the appearance of the white coat color. This study reveals the genetic mechanism of the white coat color in sika deer and supplies data as a reference for breeding white ornamental sika deer.

## 1. Introduction

Sika deer (*Cervus nippon*) is an important livestock in China that has great economic value, as most parts of its body can be used in the field of medicine and health products. The sika deer has a unique coat color phenotype that is typically brown in the summer, with white spots on the sides of the body and black backs of the ears. There are also some individuals that have a black dorsal line running from the head and neck to the tail. As winter arrives, the coat color deepens to gray, and the white spots become blurred. It is difficult to find a sika deer with any other coat color, especially white, despite the large number of sika deer in the world, both wild and captive. The white sika deer has been a symbol of good luck since ancient times in China and has great ornamental value. Precisely because of this rarity, it is very difficult to obtain samples of white sika deer to figure out why they appear; thus, the genetic mechanism of coat color in sika deer still remains unclear.

In contrast, about 5% of German red deer (*Cervus canadensis*) have white fur. Gerald found that in the 291st amino acid of the TYR (tyrosinase) protein, glycine was replaced by arginine in these white red deer through genome-wide resequencing [1]. TYR is a key synthase involved in pigment synthesis, which can catalyze the oxidation of tyrosine to form dopa and dopaquinone, as well as participate in the subsequent reactions of pigment synthesis. Although Gerald [1] confirmed that the causes of these white red deer appearing in the German red deer population were related to TYR mutation, verifying the causes of the white phenotype’s appearance in other red deer populations is still needed. Studies show that the *MC1R* (melanocortin 1 receptor) gene may regulate coat color in the white fallow deer [2], and that mutation of this gene (NM_174108.2:c.143T>C) results in an amino-acid exchange in the MC1R protein (NP_776533.1:p.L48P). MC1R is a G-protein-coupled receptor of melanocytes, which is involved in regulating the expression of MITF (microphthalmia-related transcription factor) by binding with the ligand α-MSH(α-melanocyte-stimulating hormone)or ASIP (application-specific instruction set processor). MITF can regulate the synthesis of pigment synthase and membrane transporters in a pigment synthesis reaction, ultimately affecting melanin synthesis [3].

There is a coat color trait called the roan phenotype, which is related toSCF in some ruminants, such as cattle and sheep [4,5]. Functional mutations in the *SCF* gene can cause the roan phenotype in ruminants and melanocyte defects in mice [6]. Individuals with the roan phenotype are usually heterozygous in genotype, while homozygous individuals have a white color covering their fur. Sheep with the homozygous genotype die at the newborn stage due to anemia and digestive disorders [7], and very few homozygous cows have abnormalities in the reproductive tract [8]. Mutations in this gene can also cause human familial progressive hyper- and hypopigmentation (FPHH), a condition in which people who carry the mutated *SCF* gene have not been found in homozygous mutations. Histology reveals fewer pigment granules in the skin in light-colored spots and more pigment granules in darker spots [9]. The *SCF* gene encodes a transmembrane protein that is synthesized and secreted by keratinocytes. It has two isomers, a soluble isomer and a membrane-bound isomer. The soluble isomer is mainly involved in the synthesis of melanin in melanocytes, while the membrane-bound isomer is mainly involved in the development and migration of melanocytes. Both isomers act by binding with the KIT (receptor tyrosine kinase) [10,11]. Soluble isomer precursors are transported to the cell membrane and cut by chymotrypsin [12], before being released into the extracellular space, while transmembrane isomers remain on the membrane because they do not encode hydrolysis sites [13]. Some external factors (ultraviolet B) and internal factors (inflammatory reaction) can stimulate keratinocytes to produce and secrete more SCF [14]. Stimulated by ultraviolet B or an inflammatory reaction, keratinocytes produce MIF (macrophage migration-inhibitory factor ) and induce cells to synthesize SCF. SCF increases the size and number of melanocyte dendrites, increases skin pigmentation [15,16], and darkens skin color.

In contrast to normal sika deer, we found a 1 year old male sika deer with a white coat color. It is worth noting that this sika deer’s eyes were black (Figure 1(C1)), which is similar to the coat color phenotype of black-eyed white mice [17]. Some of its family members had white spots on their faces (Figure 1(C2)), while the remainder had a normal phenotype. We speculated that a gene mutation led to the change to a white coat color, with some sika family members potentially carrying mutant genes.

In this study, we used the high-throughput sequencing technique to resequence the whole genome of the white sika deer. Then, the resequencing data were analyzed, and the information on the difference was screened by establishing the candidate gene group of coat color. Skin tissue sections of the sika deer were created, and the white sika deer families were analyzed for genotype. Our aim was to investigate the genetic mechanism of white sika deer, and to provide molecular data for white sika deer breeding.

## 2. Materials and Methods

### 2.1. Sample Collection and DNA Extraction

Blood and tissue samples of the white sika deer and its family members were taken from a sika deer farm in Nongan, Jilin Province, other samples were taken from the Jilin sika deer germplasm conservation farm of the Institute of Special Animal and Plant Sciences, Chinese Academy of Agricultural Sciences (CAAS). All animals involved were treated in accordance with The Guide for Care and Use of Laboratory Animals formulated by the Ministry of Agriculture and Rural Affairs of the People’s Republic of China. All protocols were approved by the Institutional Animal Care and Use Professional Committee of Institute of Special Animal and Plant Sciences of CAAS. Blood DNA was extracted using the TIANamp Blood DNA KIT (TianGen, Beijing, China).

### 2.2. Whole-Genome Resequencing and Sample Comparison

The whole-genome sequence of the white sika deer was sequenced by Illumina NovoSeq PE150 (Glbizzia, Beijing, China). The raw reads sequenced were filtered using fastp [17] to remove low-quality reads including (1) reads with adaptor contamination, (2) paired reads when uncertain nucleotides constituted more than 10% of either read, and (3) paired reads when low-quality nucleotides (base quality less than 5) constituted more than 50% of the read.

The clean data were matched to the Maitreya deer reference genome (GWHANOY00000000) by BWA software [18] (parameter: mem -t 10 -M) [19], and the results were matched with SAMTOOLS [20] to remove duplicates (parameter: rmdup).

### 2.3. Variation Detection and Annotation

The GATK 4.2.2.0 HaplotypeCaller module was used to perform variation detection on the bam files after MarkDuplicates to obtain gvcf. The GenotypeGVCFs module was used to assign genotypes, and the SelectVariants module was used to split SNPs and InDels(insertions/deletions ); CNV (copy number variant) calling andSV calling were performed using CNVnator v0.4.1 and lumpy v0.3.1-3, respectively, and SURVIVOR 1.0.7 was used to merge those CNVs and SVs. The ANNOVAR software was used to annotate these variations.

### 2.4. Establishment of Candidate Genome

By referring to the genetic variation analysis methods of albinism [21] and white tigers [22], a candidate genome of coat color was established, and the differential information between genes of the white sika deer and of the cluster located in exonic regions was screened. The candidate genome was established according to the gene information provided by the website of the IFPCs (International Federation of Pigment Cell Societies) (http://www.ifpcs.org/colorgenes/, accessed on 1 December 2021). We selected 92 coat color genes (Appendix A) from this website.

### 2.5. Screening of Coat Color Gene Variation

The resequencing data of 152 sika deer (unpublished) were selected as the control. AF (allele frequency) values of white sika deer and control were calculated, and SNPs with the absolute value of AF difference between the two groups ≥0.8 were selected. The bedtools v2.30.0 intersect module was used to extract unique InDels, CNVs, and SVs in the two groups with <30% overlap. The difference information located in the exon region of candidate genes in these data was extracted.

### 2.6. Preparation and Staining of Paraffin Skin Tissue Sections

#### 2.6.1. Skin Tissue Sectioning

The antler was soaked in 4% paraformaldehyde for 24 h, and then cut into 0.5 cm × 0.5 cm segments of skin tissue, which were rinsed under running water for 20 min, dehydrated using an ethanol gradient, and embedded in paraffin wax. The wax pieces were cut using a slicer, absorbed by a glass slide, and baked in the oven overnight.

#### 2.6.2. HE Staining

Slices were soaked in xylene for 30 min, then soaked in 100% ethanol, 100% ethanol, 95% ethanol, 90% ethanol, 80% ethanol, and 70% ethanol for 2 min, rinsed with running water for 15 min, soaked in Beyotime (C0105) for 2 min to stain the nuclei, and rinsed with running water for 5 min. The slices were dehydrated to 95% using an ethanol gradient, and the soaking time was 2 min. Some slices were soaked in 95% eosin (Beyotime, Shanghai, China) for 2 min to stain the cytoplasm, while the remainder went directly to the next step, involving dehydration with 100% ethanol and soaking for 2 min. The dyed slices were soaked in xylene for 30 min, during which the solution was changed once. The slices were sealed with neutral gum and baked in an oven overnight at 37 °C.

#### 2.6.3. L-DOPA Staining

First, 0.1% L-DOPA (Medchemexpress, Tianjin, China) solution was prepared with PBS buffer; then, skin tissue blocks were immersed in this solution and incubated in the incubator for 24 h at 37 °C. The incubated tissue was rinsed with running water for 20 min, before performing the steps described in Section 2.6.1 and Section 2.6.2 to obtain slices. During the operation, only eosin staining was used, whereas hematoxylin staining was not used. The prepared tissue sections were observed under a microscope and photographed.

### 2.7. Phenotypic Genotyping and Sanger Sequencing

The family atlas of the white sika family was plotted on the basis of genealogical records from the deer farm. The Primer 5.0 software was used to design specific SCF gene primers. Primer SCF-1 could only amplify mutated gene sequence fragments, while primers SCF-2, SCF-3, and SCF-4 could only amplify normal gene sequence fragments. The amplification reaction system was set to 25 μL: 11.5 μL of Mix (Cwbio, jiangsu, China), 11.5 μL of ddH_2_O, 0.5 μL each of the upstream and downstream primers, and 1 μL of DNA template. Reaction conditions were as follows: pre-denaturation at 95 °C for 5 min; denaturation at 30 cycles of 95 °C for 10 s (see Appendix A for annealing temperature); extension at 72 °C for 30 s; holding at 4 °C for 5 min.

After amplification, the standard bands were electrophoresed on a 2% gel, and standard bands were used with DL1000 DNA marker (Takara, Beijing, China). The presence or absence of electrophoretic bands were observed, the family members were genotyped, and the inheritance mechanism of the mutated gene was analyzed in relation to the phenotype and genotype. The samples amplified from SCF-1 were used for Sanger sequencing, and the MEGA 7.0 software was used to compare whether there were differences between sequenced fragments and the normal *SCF* gene fragment sequence, which was extracted from the reference genome of sika deer.

## 3. Results

### 3.1. Resequencing Data and Genetic Variation

The sequencing results were filtered for quality control, yielding 98.4 Gb of clean data, with 99.19% genome coverage and 39.06× sequencing depth. After allele frequency analysis, 780,147 SNP loci, 83,287 InDels, 4380 SVs, and 448 CNVs were screened. A total of one SV and 19 SNPs were located on the exon sequence of the coat color candidate gene, including five nonsynonymous mutations and 14 synonymous mutations in SNPs (Table 1). The SV was located within the SCF gene (chr3:16,444,201–16,454,300), which lost a 10.099 kb DNA fragment. The nonsynonymous mutations were *TYRP1* (c.C137T:p S46F), *LYST* (c.C1667T:p.S556L), *ICAT* (c.C268G:p.P90A), *WRN* (c.T1529C:p.I510T), and *EPG5* (c.A4762G:p.S1588G), of which the first three were albino pathogenic genes [23].

### 3.2. Staining Results for Tissue Sections

The results of HE staining (Figure 2(A)) revealed that black and yellow pigment particles were present in the follicles of normal sika skin tissues, whereas no pigment granules were found in the follicles of white sika skin tissues. After the skin tissue was incubated with L-DOPA (Figure 2(B)), the number of pigment particles in the skin tissue of normal sika increased, but there was no obvious change in the skin tissue of white sika.

### 3.3. Analysis of SCF Genotype and Phenotype in White Sika Deer Family

In the family atlas of white sika deer (Figure 3a), the maternal (F), paternal (D), and grandfather’s (C) faces all had white spots, whereas the remainder (E, G, and H) had normal phenotypes. The electrophoresis chart (Figure 3(b1)) showed that the control (A) had primer amplification bands of SCF-2, SCF-3, and SCF-4, as well as of SCF-1, but the size of the bands did not match; thus, SCF-1 could not be amplified normally. The information on primer positions is shown in the Figure 3 (b2). The mutant group (B, white sika deer) had SCF-1 amplification bands, but no SCF-2, SCF-3, and SCF-4 amplification bands. According to the family map and genotype analysis (Table 2), it was found that the individual genotypes of C, D, and F were heterozygous (SCF^789^/SCF^1-9^), and their coat color phenotypes were normal except for white patches on their faces. Genotypes E, G, and H were consistent with that of A (SCF^1-9^/SCF^1-9^), and the coat color phenotype was normal. The genotype of the white sika deer (B) was homozygous (SCF^789^/SCF^789^), and its coat color phenotype was snow white.

### 3.4. Sanger Sequencing

SCF-1 (B–D, F) was used to represent the sequencing results of samples B, C, D, and G (Figure 3(b1)), while SCF-1 (normal) was used to represent SCF gene fragments. The results of comparison with MEGA5.0 software showed that the SCF-1 (B–D, F) DNA fragments completely matched with those at both ends of SCF-1 (normal), with about 10 kb of DNA being lost in the middle, consistent with the resequencing results.

## 4. Discussion

Due to the small number of white sika deer, a significant amount of differential information was screened out from the resequencing data as it was not conducive to subsequent analysis. Therefore, we personalized the screened data after AF(allele frequency) analysis by establishing the candidate genome of coat color, which was composed of coat color genes related to a white phenotype and animal albinism in other mammals. We then selectively screened out the required difference regions. However, we only selected the SV and five nonsynonymous mutations located in the exon sequence of the candidate gene, representing the coat color genes that may be related to the white phenotype of sika deer. The exon is an important gene coding sequence in animals, and changes in coat color and related diseases in mammals are mostly caused by mutation of the gene exon region [21]. In addition, synonymous mutation in the exon would not lead to transcription and translation changes, thus not affecting the functional activity of proteins. To summarize, we directly excluded synonymous mutation from the differential information.

As the pigment precursor, L-DOPA can be formed into melanin through a subsequent pigment synthesis reaction [24]. The melanosome of melanocytes is the main site of melanin synthesis reactions, which is a membranous organelle [14]. With the accumulation of melanin, melanosome turns black and spreads to surrounding cells [25]. These pigment particles in skin tissue were observed under a microscope. However, there were no pigment particles in the skin tissue of white sika deer, and there was no reaction to L-DOPA, indicating that the pigment synthesis process was completely blocked. However, the white sika deer had black eyes, showing that pigment particles still existed in the eyes, with the pigment synthesis process not being blocked. The sources of melanocytes in mammalian skin and eyes are different. Melanocytes in skin are derived from the neural crest, whereas melanocytes in eyes (melanocyte epithelium) are mainly derived from the neuroectoderm. Although the two melanocytes are differentiated from different cells, both synthesize and store melanin in the cytoplasm [26], and there are no differences in the melanogenesis process. This suggests that the process of pigment synthesis in the skin of white sika is disturbed by other factors. In addition, there are two main ways to synthesize mammalian pigments. The first pathway produces eumelanin, while the second produces brownish melanin [27]. The research showed that the mutations of *TYRP1*, *LYST* [28], *ICAT*, *WRN*, and *EPG5* [29] would lead to hypopigmentation in the skin, coat, and eyes. Although the amount of pigment synthesis in melanocytes is reduced, the synthesis process is not completely blocked. The mutation of a single gene only affects one pathway of pigment synthesis, with no obvious effect on the other pathway. This may be the reason why another clinical manifestation of albinism involved skin, hair, and eye pigment dilution rather than complete albinism, except for albinism caused by *TYR* gene mutation. The *SCF* gene not only regulates pigment synthesis, but also participates in the development and migration of melanocytes. The structural variation in the *SCF* gene had a strong correlation with the coat color changes of white sika deer.

In the process of designing primers, we found that the mutation position of the sika deer *SCF* gene included exons 7, 8, and 9. The exon sequence of the *SCF* gene was extracted from the sika deer genome, and the structure of the SCF protein was predicted using online software (https://services.healthtech.dtu.dk/service.php?TMHMM-2.0, accessed on 20 January 2023) after sequence translation (Appendix A). The mutation position was located in the transmembrane region and part of the intramembrane region of the *SCF* protein. Surprisingly, a similar *SCF* gene mutation was identified in SLD (Steel–Dickie) mice [30]. This kind of mouse has black eyes and a white coat color, and its features are also caused by structural variation in the *SCF* gene. Mutation caused a deletion of 4.0 kb in the *SCF* gene sequence, resulting in keratinocytes only encoding a soluble isomer lacking transmembrane and cytoplasmic domains [31]. The membrane-bound isoform cannot be encoded after mutations in the *SCF* gene, which plays a major role in melanocyte development and migration. This can guide cells to move along the correct route by mediating adhesion [30]. Although the mutant *SCF* gene may still retain the ability to encode soluble isomers, neural crest cells that did not differentiate into melanocytes during embryonic development cannot transfer to the epidermal basal layer and follicles due to the migration disorder of melanocytes. Although these SNPs can reflect the polymorphism of genes related to the sika deer’s coat color to a certain degree, this polymorphism has little to do with the extreme change in coat color since the change in coat color tends to be triggered by the change in protein function. The skin tissue of the body loses the ability to synthesize melanin because of the lack of melanocytes.

Genotyping further confirmed the correlation between the white phenotype of sika deer and the *SCF* gene. Among the four primers used for genotyping, SCF-1 was designed on the basis of the mutant *SCF* gene sequence, which could amplify about 294 bp bands at both ends of the deleted fragment. The normal *SCF* gene could not amplify SCF-1 because it retained the deleted fragment of 10.099 kb. SCF-2, SCF-3, and SCF-4 were located in the deleted fragments, and the mutant *SCF* gene could not amplify the bands because of the loss of the deleted fragments. Genotyping showed that the male parent, female parent, and grandfather of white sika deer with white spots on their faces all carried the mutated SCF gene, and it was indicated that the white spots on their faces were also related to SCF gene mutation. The DNA of white sika deer only amplified the band of SCF-1; hence, it did not carry the normal SCF gene. The genotype and phenotype of the family members were completely matched. Studies have shown that the SCF gene also controls the white phenotype in Belgian blue cattle and short-horned cattle. The coding sequence of the SCF gene in both kinds of cattle has a base substitution at 654 bp, which leads to the substitution of Ala by Asp [9]. Belgian blue cattle have three coat color phenotypes: black spots, blue and white, and blue as a mixture of black and white. The coat color of short-horned cattle has similar phenotypes: black, black–red, and white [4]. The coat color phenotype of homozygous mutant Belgian blue cattle and short-horned cattle is all white, and whereas heterozygous cattle have blue or black–red coats. Similarly, the homozygous mutation phenotype in sika deer is white, while the heterozygous individual phenotype features white spots on the face; the inheritance mode of coat color is also single-gene dominant inheritance.

## 5. Conclusions

Through experiments, we found that the white phenotype of sika deer is related to structural variation in the SCF gene. A 10.099 kb fragment comprising the seventh, eighth, and ninth exons of the SCF gene was lost in white sika deer. This led to structural defects and the loss of some functions of the protein coded by the SCF gene, affecting the development and migration of melanocytes. The emergence of white sika deer was caused by a defect in melanocytes in the skin tissue. The inheritance mode of this coat color trait is single-gene dominant inheritance. The phenotype of dominant homozygous individuals is white, while heterozygous individuals have white patches on their faces.

## Figures and Tables

**Figure 1 genes-14-01035-f001:**
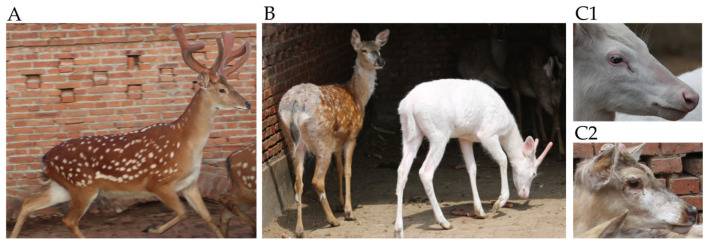
(**A**) Normal sika deer. (**B**) Sika deer with white spots (left) and white coat color (right). (**C1**) Head characteristics of sika deer with white coat color. (**C2**) Head characteristics of sika deer with white spots.

**Figure 2 genes-14-01035-f002:**
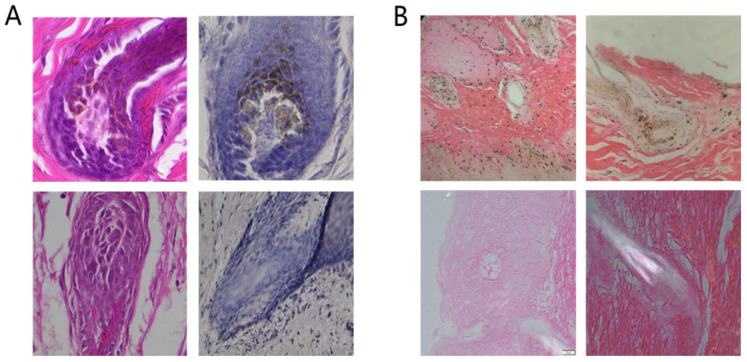
(**A**): HE staining of coat follicle sections: normal (upper); white (below). (**B**) L-DOPA staining: normal (upper); white (bottom).

**Figure 3 genes-14-01035-f003:**
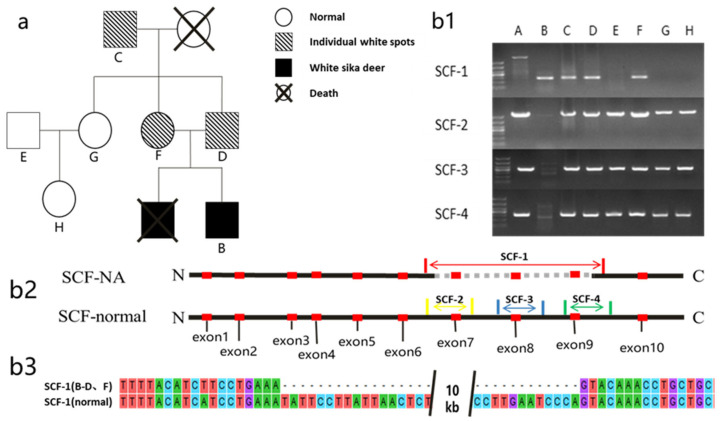
(**a**) Genetic atlas of the white sika deer family. (**b1**) Electropherogram: A (control); B–H (individual deer corresponding to Figure 3a). Primer amplification positions corresponded to Figure 3(b2). (**b2**) Location of exons and amplification regions of the SCF gene, along with the mutation regions (dashed areas), in sika deer. (**b3**) Sequence comparison.

**Table 1 genes-14-01035-t001:** Mutant genes information.

Genes	AF	Mutation Position	Chromosome	Type
CIITA	0.819079	exon10:c.C1209T:p.S403S	MHL_chr10	Synonymous SNV
CIITA	0.8125	exon10:c.G1644T:p.S548S	MHL_chr10	Synonymous SNV
AP3B1	0.9471831	exon23:c.A2754G:p.Q918Q	MHL_chr12	Synonymous SNV
ICAT	0.822368	exon2:c.C268G:p.P90A	MHL_chr14	Nonsynonymous SNV
LYST	0.832237	exon9:c.C1667T:p.S556L	MHL_chr15	Nonsynonymous SNV
HPS3	0.834507	exon9:c.G1590A:p.A530A	MHL_chr19	Synonymous SNV
GSK3-β	0.835526	exon10:c.G1047A:p.L349L	MHL_chr19	Synonymous SNV
DKK3	0.888158	exon4:c.G510A:p.P170P	MHL_chr1	Synonymous SNV
DKK3	0.9205298	exon4:c.C513T:p.C171C	MHL_chr1	Synonymous SNV
EPG5	0.9276316	exon11:c.C2127T:p.P709P	MHL_chr27	Synonymous SNV
EPG5	0.9337748	exon27:c.A4762G:p.S1588G	MHL_chr27	Nonsynonymous SNV
EPG5	0.807947	exon29:c.C5040T:p.D1680D	MHL_chr27	Synonymous SNV
EPG5	0.9342105	exon32:c.T5643C:p.V1881V	MHL_chr27	Synonymous SNV
TYRP1	0.838816	exon1:c.C137T:p.S46F	MHL_chr29	Nonsynonymous SNV
WRN	0.809211	exon12:c.T1529C:p.I510T	MHL_chr32	Nonsynonymous SNV
SLC38A8	0.894737	exon1:c.C24T:p.G8G	MHL_chr4	Synonymous SNV
SLC38A8	0.888158	exon6:c.C696T:p.H232H	MHL_chr4	Synonymous SNV
HPS4	0.9212329	exon9:c.G813A:p.A271A	MHL_chr5	Synonymous SNV
VPS33A	0.886986	exon6:c.G735T:p.T245T	MHL_chr5	Synonymous SNV
SCF	-	c.16444201-16454300	MHL_chr3	CNV

**Table 2 genes-14-01035-t002:** Genotype and phenotype distribution.

Phenotype	Genotype	Individuals	Amount
White	SCF^789^/SCF^789^	B	1
White spots	SCF^789^/SCF^1–9^	C, D, and F	3
Normal	SCF^1–9^/SCF^1–9^	E, G, and H	3

Note: Individual numbering is consistent with Figure 3a.

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
