# Peer review of "Phenotype of White Sika Deer Due to SCF Gene Structural Variation"

_genes, 2023, doi:10.3390/genes14051035_

Round 1
Reviewer 1 Report
Dear authors
This is a very interesting study, you have used the appropriate methodology but there are some concerns for the manuscript
Line 29: "It has a singular and genetically stable coat colour phenotype". There is no reference on this and it should be rephrased
Line 23-31: Some animals have a different phenotype. So is there a singular phenotype or not? I cannot understand
Line 34-35: White sika deer have been a symbol of 34 good fortune in China since ancient times. It is believed that seeing a white sika deer will 35 bring good luck. Please remove these lines. This is not scientific
Line 42: residue instead of site
Lines 64-69 : please rephrase.
Line 71: what do you mean by "differential loci"?
Materials and methods should be organized better and presented in a more clear way.
Discussion: Do you have results on belgian blue? I dont understand
Also the discussion is organized around different species with little references in your species. You should focus primarily on your results.
Furthermore, the discussion is rather small, you should elaborate more on your results
Author Response
Dear Reviewer,
Thank you for your patient examination of the manuscript. For some of the questions you raised, I have revised them and written the handling situation in the attachment. Please see the attachment.
Kind regards,
authors

Reviewer 2 Report
Dear authors,
I liked very much of your work, from the study design to the data analysis and interpretation. Coat colour phenotype is a topic of great interest worldwide, and the results provided in this study for sika deer is one more evidence of the implication of SCF gene on determining the white type.
I would like to point out only two thinks: first, in the discussion section you should start stressing your results and thereafter integrate them with known knowledge from other species, and not the reverse. It is hard to understand the first paragraph since you describe the results from other studies and species. Also, the authors should avoid discuss the obtained results in the Result section. For instance, lines 179-183, but there are others. I suggest to compile all the discussion parts and re-write all the discussion section accordingly.
Second, a better context should be provide for the staining sections. I do not understand how the authors discard the implication of the other candidate genes base on staining. Maybe, because it is not clearly explained.
Best regards,
Author Response

(The authors gave the same response as above.)

Reviewer 3 Report
The authors have taken a good effort to present the findings of their study. Though many experts have explained the difference in coat color among animals theoretically, reporting such quantifiable data is always appreciable. Apart from a few typing errors (which I suppose would be dealt during proof reading), I do not have any major concerns apart from a just few suggestions as listed below:
1. The text between lines 70-81 gives a quick overview of the main findings of the study. This however would not be ideal to be indicated in the introduction. Hence, I suggest the authors delete this portion from the introduction. They could insert it after making suitable modifications either in the results/discussion/conclusion if they find it fit.
2. Additionally, the introduction section would need a minor edit to end it up by stating the hypothesis/objective of the study.
3. Also, kindly mention the white sika deer’s age and sex (either in the introduction or materials and methods)
4. Figure 1 have not been cited anywhere in the text
5. Likewise line 177-183: this matter in the result section is ideally a content to be in the ‘discussion’. Since this manuscript has distinguishable ‘results’ and ‘discussion’ sections, its ideal to discuss the results of the study ONLY in the ‘discussion’ section and not in the ‘results’. Kindly ensure this throughout the manuscript.
Author Response

(The authors gave the same response as above.)

Round 2
Reviewer 1 Report
The manuscript is really improved and now it is appropriate for publication (after some minor changes in language)